# NON-AUTOREGRESSIVE DIALOG STATE TRACKING

**Hung Le**[‡][*]**Richard Socher**[†]**, Steven C.H. Hoi**[†]
† Salesforce Research
{rsocher,shoi}@salesforce.com
‡ Singapore Management University
hungle.2018@phdcs.smu.edu.sg

## ABSTRACT

Recent efforts in Dialogue State Tracking (DST) for task-oriented dialogues have progressed toward open-vocabulary or generation-based approaches where the models can generate slot value candidates from the dialogue history itself. These approaches have shown good performance gain, especially in complicated dialogue domains with dynamic slot values. However, they fall short in two aspects: (1) they do not allow models to explicitly learn signals across domains and slots to detect potential dependencies among *(domain, slot)* pairs; and (2) existing models follow auto-regressive approaches which incur high time cost when the dialogue evolves over multiple domains and multiple turns. In this paper, we propose a novel framework of Non-Autoregressive Dialog State Tracking (NADST) which can factor in potential dependencies among domains and slots to optimize the models towards better prediction of dialogue states as a complete set rather than separate slots. In particular, the non-autoregressive nature of our method not only enables decoding in parallel to significantly reduce the latency of DST for real-time dialogue response generation, but also detect dependencies among slots at token level in addition to slot and domain level. Our empirical results show that our model achieves the state-of-the-art joint accuracy across all domains on the MultiWOZ 2.1 corpus, and the latency of our model is an order of magnitude lower than the previous state of the art as the dialogue history extends over time.

## 1 INTRODUCTION

In task-oriented dialogues, a dialogue agent is required to assist humans for one or many tasks such as finding a restaurant and booking a hotel. As a sample dialogue shown in Table 1, each user utterance typically contains important information identified as slots related to a dialogue domain such as *attraction-area* and *train-day*. A crucial part of a task-oriented dialogue system is Dialogue State Tracking (DST), which aims to identify user goals expressed during a conversation in the form of dialogue states. A dialogue state consists of a set of *(slot, value)* pairs e.g. *(attraction-area, centre)* and *(train-day, tuesday)*. Existing DST models can be categorized into two types: fixed- and open-vocabulary. Fixed vocabulary models assume known slot ontology and generate a score for each candidate of *(slot,value)* (Ramadan et al., 2018; Lee et al., 2019). Recent approaches propose open-vocabulary models that can generate the candidates, especially for slots such as entity names and time, from the dialogue history (Lei et al., 2018; Wu et al., 2019).

Most open-vocabulary DST models rely on autoregressive encoders and decoders, which encode dialogue history sequentially and generate token $t_i$ of individual slot value one by one conditioned on all previously generated tokens $t_{[1:i-1]}$. For downstream tasks of DST that emphasize on low latency (e.g. generating real-time dialogue responses), auto-regressive approaches incur expensive time cost as the ongoing dialogues become more complex. The time cost is caused by two major components: length of dialogue history i.e. number of turns, and length of slot values. For complex dialogues extended over many turns and multiple domains, the time cost will increase significantly in both encoding and decoding phases.

Similar problems can be seen in the field of Neural Machine Translation (NMT) research where a long piece of text is translated from one language to another. Recent work has tried to improve the

---

[*]All work was done while the first author was a research intern at Salesforce Research Asia.

| Human: | i want to visit a theater in the center of town . |
|---|---|
| **Dialog State:** | (attraction-area, centre), (attraction-type, theatre) |
| System: | there are 4 matches . i do not have any info on the fees . do you have any other preferences ? |
| Human: | no other preferences , i just want to be sure to get the phone number of whichever theatre we pick . |
| **Dialog State:** | (attraction-area, centre), (attraction-type, theatre) |
| System: | i recommend the cambridge corn exchange there phone number is 01223357851 . is there anything else i can help you with ? |
| Human: | yes , i am looking for a tuesday train . |
| **Dialog State:** | (attraction-area, centre), (attraction-name, the cambridge corn exchange) , (attraction-type, theatre), (train-day, tuesday) |
| System: | where will you be departing from and what s your destination ? |
| Human: | from cambridge to london liverpool street . |
| **Dialog State:** | (attraction-area, centre), (attraction-name, the cambridge corn exchange) , (attraction-type, theatre), (train-day, tuesday), (train-departure, cambridge), (train-destination, london liverpool street) |

Table 1: A sample task-oriented dialogue with annotated dialogue states after each user turn. The dialogue states in red and blue denote slots from the *attraction* domain and *train* domain respectively. Slot values are expressed in user and system utterances (highlighted by underlined text).

latency in NMT by using neural network architectures such as convolution (Krizhevsky et al., 2012) and attention (Luong et al., 2015). Several non- and semi-autoregressive approaches aim to generate tokens of the target language independently (Gu et al., 2018; Lee et al., 2018; Kaiser et al., 2018). Motivated by this line of research, we thus propose a non-autoregressive approach to minimize the time cost of DST models without a negative impact on the model performance.

We adopt the concept of *fertility* proposed by Gu et al. (2018). Fertility denotes the number of times each input token is copied to form a sequence as the input to the decoder for non-autoregressive decoding. We first reconstruct dialogue state as a sequence of concatenated slot values. The result sequence contains the inherent structured representation in which we can apply the fertility concept. The structure is defined by the boundaries of individual slot values. These boundaries can be easily obtained from dialogue state itself by simply measuring number of the tokens of individual slots. Our model includes a two-stage decoding process: (1) the first decoder learns relevant signals from the input dialogue history and generates a fertility for each input slot representation; and (2) the predicted fertility is used to form a structured sequence which consists of multiple sub-sequences, each represented as (*slot token×slot fertility*). The result sequence is used as input to the second decoder to generate all the tokens of the target dialogue state at once.

In addition to being non-autoregressive, our models explicitly consider dependencies at both slot level and token level. Most of existing DST models assume independence among slots in dialogue states without explicitly considering potential signals across the slots (Wu et al., 2019; Lee et al., 2019; Goel et al., 2019; Gao et al., 2019). However, we hypothesize that it is not true in many cases. For example, a good DST model should detect the relation that *train_departure* should not have the same value as *train_destination* (example in Table 1). Other cases include time-related pairs such as (*taxi_arriveBy*, *taxi_leaveAt*) and cross-domain pairs such as (*hotel_area*, *attraction_area*). Our proposed approach considers all possible signals across all domains and slots to generate a dialogue state as a set. Our approach directly optimizes towards the DST evaluation metric *Joint Accuracy* (Henderson et al., 2014b), which measures accuracy at state (set of slots) level rather than slot level.

Our contributions in this work include: (1) we propose a novel framework of Non-Autoregressive Dialog State Tracking (NADST), which explicitly learns inter-dependencies across slots for decoding dialogue states as a complete set rather than individual slots; (2) we propose a non-autoregressive decoding scheme, which not only enjoys low latency for real-time dialogues, but also allows to capture dependencies at token level in addition to slot level; (3) we achieve the state-of-the-art performance on the multi-domain task-oriented dialogue dataset "MultiWOZ 2.1" (Budzianowski et al., 2018; Eric et al., 2019) while significantly reducing the inference latency by an order of magnitude; (4) we conduct extensive ablation studies in which our analysis reveals that our models can detect potential signals across slots and dialogue domains to generate more correct "sets" of slots for DST.

## 2 RELATED WORK

Our work is related to two research areas: dialogue state tracking and non-autoregressive decoding.

### 2.1 DIALOGUE STATE TRACKING

Dialogue State Tracking (DST) is an important component in task-oriented dialogues, especially for dialogues with complex domains that require fine-grained tracking of relevant slots. Traditionally,

DST is coupled with Natural Language Understanding (NLU). NLU output as tagged user utterances is input to DST models to update the dialogue states turn by turn (Kurata et al., 2016; Shi et al., 2016; Rastogi et al., 2017). Recent approaches combine NLU and DST to reduce the credit assignment problem and remove the need for NLU (Mrkšić et al., 2017; Xu & Hu, 2018; Zhong et al., 2018). Within this body of research, Goel et al. (2019) differentiates two DST approaches: fixed- and open-vocabulary. Fixed-vocabulary approaches are usually retrieval-based methods in which all candidate pairs of *(slot, value)* from a given slot ontology are considered and the models predict a probability score for each pair (Henderson et al., 2014c; Ramadan et al., 2018; Lee et al., 2019). Recent work has moved towards open-vocabulary approaches that can generate the candidates based on input text i.e. dialogue history (Lei et al., 2018; Gao et al., 2019; Wu et al., 2019). Our work is more related to these models, but different from most of the current work, we explicitly consider dependencies among slots and domains to decode dialogue state as a complete set.

## 2.2 Non-autoregressive Decoding

Most of prior work in non- or semi-autoregressive decoding methods are used for NMT to address the need for fast translation. Schwenk (2012) proposes to estimate the translation model probabilities of a phase-based NMT system. Libovickỳ & Helcl (2018) formulates the decoding process as a sequence labeling task by projecting source sequence into a longer sequence and applying CTC loss (Graves et al., 2006) to decode the target sequence. Wang et al. (2019) adds regularization terms to NAT models (Gu et al., 2018) to reduce translation errors such as repeated tokens and incomplete sentences. Ghazvininejad et al. (2019) uses a non-autoregressive decoder with masked attention to decode target sequences over multiple generation rounds. A common challenge in non-autoregressive NMT is the large number of sequential latent variables, e.g., fertility sequences (Gu et al., 2018) and projected target sequences (Libovickỳ & Helcl, 2018). These latent variables are used as supporting signals for non- or semi-autoregressive decoding. We reformulate dialogue state as a structured sequence with sub-sequences defined as a concatenation of slot values. This form of dialogue state can be inferred easily from the dialogue state annotation itself whereas such supervision information is not directly available in NMT. The lower semantic complexity of slot values as compared to long sentences in NMT makes it easier to adopt non-autoregressive approaches into DST. According to our review, we are the first to apply a non-autoregressive framework for generation-based DST. Our approach allows joint state tracking across slots, which results in better performance and an order of magnitude lower latency during inference.

## 3 Approach

Our NADST model is composed of three parts: encoders, fertility decoder, and state decoder, as shown in Figure 1. The input includes the dialogue history $X = (x_1, ..., x_N)$ and a sequence of applicable *(domain, slot)* pairs $X_{\text{ds}} = ((d_1, s_1), ..., (d_G, s_H))$, where $G$ and $H$ are the total numbers of domains and slots, respectively. The output is the corresponding dialogue states up to the current dialogue history. Conventionally, the output of dialogue state is denoted as tuple *(slot, value)* (or *(domain-slot, value)* for multi-domain dialogues). We reformulate the output as a concatenation of slot values $Y^{d_i, s_j}$: $Y = (Y^{d_1, s_1}, ..., Y^{d_I, s_J}) = (y_1^{d_1, s_1}, y_2^{d_1, s_1}, ..., y_1^{d_I, s_J}, y_2^{d_I, s_J}, ...)$ where $I$ and $J$ are the numbers of domains and slots in the output dialogue state, respectively.

First, the encoders use token-level embedding and positional encoding to encode the input dialogue history and *(domain, slot)* pairs into continuous representations. The encoded domains and slots are then input to stacked self-attention and feed-forward network to obtain relevant signals across dialogue history and generate a fertility $Y_f^{d_g, s_h}$ for each *(domain, slot)* pair $(d_g, s_h)$. The output of fertility decoder is defined as a sequence: $Y_{\text{fert}} = Y_f^{d_1, s_1}, ..., Y_f^{d_G, s_H}$ where $Y_f^{d_g, d_h} \in \{0, \max(\text{SlotLength})\}$. For example, for the MultiWOZ dataset in our experiments, we have $\max(\text{SlotLength}) = 9$ according to the training data. We follow (Wu et al., 2019; Gao et al., 2019) to add a slot gating mechanism as an auxiliary prediction. Each gate $g$ is restricted to 3 possible values: "none", "dontcare" and "generate". They are used to form higher-level classification signals to support fertility decoding process. The gate output is defined as a sequence: $Y_{\text{gate}} = Y_g^{d_1, s_1}, ..., Y_g^{d_G, s_H}$.

The predicted fertilities are used to form an input sequence to the state decoder for non-autoregressive decoding. The sequence includes sub-sequences of $(d_g, s_h)$ repeated by $Y_f^{d_g, s_h}$ times

and concatenated sequentially: $X_{\mathrm{ds}\times\mathrm{fert}} = ((d_1, s_1)^{Y_f^{d_1,s_1}}, ..., (d_G, s_H)^{Y_f^{d_G,s_H}})$ and $\|X_{\mathrm{ds}\times\mathrm{fert}}\| = \|Y\|$. The decoder projects this sequence through attention layers with dialogue history. During this decoding process, we maintain a memory of hidden states of dialogue history. The output from the state decoder is used as a query to attend on this memory and copy tokens from the dialogue history to generate a dialogue state.

Following Lei et al. (2018), we incorporate information from previous dialogue turns to predict current turn state by using a partially delexicalized dialogue history $X_{\mathrm{del}} = (x_{1,\mathrm{del}}, ..., x_{N,\mathrm{del}})$ as an input of the model. The dialogue history is delexicalized till the last system utterance by removing real-value tokens that match the previously decoded slot values to tokens expressed as *domain-slot*. Given a token $x_n$ and the current dialogue turn $t$, the token is delexicalized as follows:

$$x_{n,\mathrm{del}} = \mathrm{delex}(x_n) = \begin{cases} \mathrm{domain}_{\mathrm{idx}}\text{-}\mathrm{slot}_{\mathrm{idx}}, & \text{if } x_n \subset \hat{Y}_{t-1}. \\ x_n, & \text{otherwise.} \end{cases} \tag{1}$$

$$\mathrm{domain}_{\mathrm{idx}} = X_{\mathrm{ds}\times\mathrm{fert}}[\mathrm{idx}][0], \quad \mathrm{slot}_{\mathrm{idx}} = X_{\mathrm{ds}\times\mathrm{fert}}[\mathrm{idx}][1], \quad \mathrm{idx} = \mathrm{Index}(x_n, \hat{Y}_{t-1}) \tag{2}$$

For example, the user utterance "I look for a cheap hotel" is delexicalized to "I look for a *hotel_pricerange* hotel." if the slot *hotel_pricerange* is predicted as "cheap" in the previous turn. This approach makes use of the delexicalized form of dialogue history while not relying on an NLU module as we utilize the predicted state from DST model itself. In addition to the belief state, we also use the system action in the previous turn to delexicalize the dialog history in a similar manner, following prior work (Rastogi et al., 2017; Zhong et al., 2018; Goel et al., 2019).

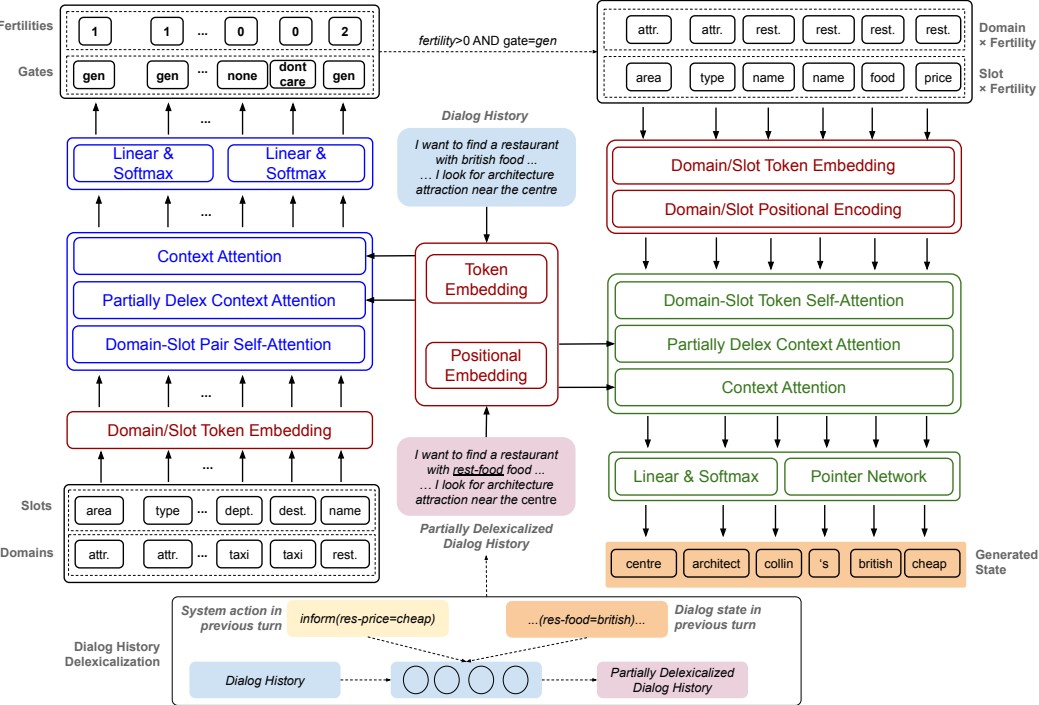

Figure 1: Our NADST has 3 key components: encoders ("red"), fertility decoder ("blue"), and state decoder ("green"). (i) *Encoders* encode sequences of dialogue history, delexicalized dialogue history, and domain and slot tokens into continuous representations; (ii) *Fertility Decoder* has 3 attention mechanisms to learn potential dependencies across *(domain, slot)* pairs in combination with dialogue history. The output is used to generate fertilities and slot gates; and (iii) *State Decoder* receives the input sequence including sub-sequences of *(domain, slot)×fertility* to decode a complete dialogue state sequence as concatenation of component slot values. For simplicity, we do not show feedforward, residual connection, and layer-normalization layers in the figure. Best viewed in color.

### 3.1 ENCODERS

An encoder is used to embed dialogue history $X$ into a sequence of continuous representations $Z = (z_1, ..., z_N) \in \mathbb{R}^{N \times d}$. Similarly, partially delexicalized dialogue history $X_{del}$ is encoded to continuous representations $Z_{\text{del}} \in \mathbb{R}^{N \times d}$. We store the encoded dialogue history $Z$ in memory which will be passed to a pointer network to copy words for dialogue state generation. This helps to address the OOV challenge as shown in (See et al., 2017; Wu et al., 2019). We also encode each *(domain, slot)* pair into continuous representation $z_{\text{ds}} \in \mathbb{R}^d$ as input to the decoders. Each vector $z_{\text{ds}}$ is used to store contextual signals for slot and fertility prediction during the decoding process.

**Context Encoder**. Context encoder includes a token-level trainable embedding layer and layer normalization (Ba et al., 2016). The encoder also includes a positional encoding layer which follows sine and cosine functions (Vaswani et al., 2017). An element-wise summation is used to combine the token-level vectors with positional encoded vectors. We share the embedding weights to embed the raw and delexicalized dialogue history. The embedding weights are also shared to encode input to both fertility decoder and state decoder. The final embedding of $X$ and $X_{del}$ is defined as:

$$Z = Z_{\text{emb}} + \text{PE}(X) \in \mathbb{R}^{N \times d} \tag{3}$$

$$Z_{\text{del}} = Z_{\text{emb,del}} + \text{PE}(X_{\text{del}}) \in \mathbb{R}^{N \times d} \tag{4}$$

**Domain and Slot Encoder**. Each *(domain, slot)* pair is encoded by using two separate embedding vectors of the corresponding domain and slot. Each domain $g$ and slot $h$ is embedded into a continuous representation $z_{d_g}$ and $z_{s_h} \in \mathbb{R}^d$. The final vector is combined by element-wise summation:

$$z_{d_g, s_h} = z_{d_g} + z_{s_h} \in \mathbb{R}^d \tag{5}$$

We share the embedding weights to embed domain and slot tokens in both fertility decoder and state decoder. However, for input to state decoder, we inject sequential information into the input $X_{\text{ds} \times \text{fert}}$ to factor in position-wise information to decode target state sequence. In summary, $X_{\text{ds}}$ and $X_{\text{ds} \times \text{fert}}$ is encoded as following:

$$Z_{\text{ds}} = Z_{\text{emb,ds}} = z_{d_1, s_1} \oplus ... \oplus z_{d_G, s_H} \tag{6}$$

$$Z_{\text{ds} \times \text{fert}} = Z_{\text{emb,ds} \times \text{fert}} + \text{PE}(X_{\text{ds} \times \text{fert}}) \tag{7}$$

$$Z_{\text{emb,ds} \times \text{fert}} = (z_{d_1, s_1})^{Y_f^{d_1, s_1}} \oplus ... \oplus (z_{d_G, s_H})^{Y_f^{d_G, s_H}} \tag{8}$$

where $\oplus$ denotes concatenation operation. Note that different from a typical decoder input in Transformer, we do not shift the input sequences to both fertility decoder and state decoder by one position as we consider non-autoregressive decoding process in both modules. Therefore, all output tokens are generated in position $i$ based on all remaining positions of the sequence i.e. $1, ..., i-1, i+1, ...\|X_{\text{ds}}\|$ in fertility decoder and $1, ..., i-1, i+1, ...\|X_{\text{ds} \times \text{fert}}\|$ in state decoder.

### 3.2 FERTILITY DECODER

Given the encoded dialogue history $Z$, delexicalized dialogue history $Z_{\text{del}}$, and *(domain,slot)* pairs $Z_{\text{ds}}$, the contextual signals are learned and passed into each $z_{\text{ds}}$ vector through a sequence of attention layers. We adopt the multi-head attention mechanism (Vaswani et al., 2017) to project the representations into multiple sub-spaces. The attention mechanism is defined as scaled dot-product attention between query $Q$, key $K$, and value $V$:

$$\text{Attention}(Q, K, V) = \text{softmax}(\frac{QK^T}{\sqrt{d_k}} V) \tag{9}$$

Each multi-head attention is followed by a position-wise feed-forward network. The feed-forward is applied to each position separately and identically. We use two linear layers with a ReLU activation in between. The fertility decoder consists of 3 attention layers, each of which learns relevant contextual signals and incorporates them into $z_{ds}$ vectors as input to the next attention layer:

$$Z_{\text{ds}}^{\text{out}} = \text{Attention}(Z_{\text{ds}}, Z_{\text{ds}}, Z_{\text{ds}}) \in \mathbb{R}^{N \times d} \tag{10}$$

$$Z_{\text{ds}}^{\text{out}} = \text{Attention}(Z_{\text{ds}}^{\text{out}}, Z_{\text{del}}, Z_{\text{del}}) \in \mathbb{R}^{N \times d} \tag{11}$$

$$Z_{\text{ds}}^{\text{out}} = \text{Attention}(Z_{\text{ds}}^{\text{out}}, Z, Z) \in \mathbb{R}^{N \times d} \tag{12}$$

For simplicity, we do not express the multi-head and feed-forward equations. We advise the reader to review Transformer network (Vaswani et al., 2017) for more detailed description. The multi-head structure has shown to obtain good performance in many NLP tasks such as NMT (Vaswani et al., 2017) and QA (Dehghani et al., 2019). By adopting this attention mechanism, we allow the models to explicitly obtain signals of potential dependencies across *(domain, slot)* pairs in the first attention layer, and contextual dependencies in the subsequent attention layers. Adding the delexicalized dialogue history as input can provide important contextual signals as the models can learn the mapping between real-value tokens and generalized *domain-slot* tokens. To further improve the model capability to capture these dependencies, we repeat the attention sequence for $T_{\text{fert}}$ times with $Z_{\text{ds}}$. In an attention step $t$, the output from the previous attention layer $t-1$ is used as input to current layer to compute $Z_{\text{ds}}^t$. The output in the last attention layer $Z_{\text{ds}}^{T_{\text{fert}}}$ is passed to two independent linear transformations to predict fertilities and gates:

$$P^{\text{gate}} = \text{softmax}(W_{\text{gate}}Z_{\text{ds}}^{T_{\text{fert}}}) \tag{13}$$

$$P^{\text{fert}} = \text{softmax}(W_{\text{fert}}Z_{\text{ds}}^{T_{\text{fert}}}) \tag{14}$$

where $W_{\text{gate}} \in \mathbb{R}^{d \times 3}$ and $W_{\text{fert}} \in \mathbb{R}^{d \times 10}$. We use the standard cross-entropy loss to train the prediction of gates and fertilities:

$$\mathcal{L}_{\text{gate}} = \sum_{d_g, s_h} -\log(P^{\text{gate}}(Y_g^{d_g, s_h})) \tag{15}$$

$$\mathcal{L}_{\text{fert}} = \sum_{d_g, s_h} -\log(P^{\text{fert}}(Y_f^{d_g, s_h})) \tag{16}$$

## 3.3 State Decoder

Given the generated gates and fertilities, we form the input sequence $X_{\text{ds} \times \text{fert}}$. We filter out any *(domain, slot)* pairs that have gate either as "none" or "dontcare". Given the encoded input $Z_{\text{ds} \times \text{fert}}$, we apply a similar attention sequence as used in the fertility decoder to incorporate contextual signals into each $z_{\text{ds} \times \text{fert}}$ vector. The dependencies are captured at the token level in this decoding stage rather than at domain/slot higher level as in the fertility decoder. After repeating the attention sequence for $T_{\text{state}}$ times, the final output $Z_{\text{ds} \times \text{fert}}^{T_{\text{state}}}$ is used to predict the state in the following:

$$P_{\text{vocab}}^{\text{state}} = \text{softmax}(W_{\text{state}}Z_{\text{ds} \times \text{fert}}^{T_{\text{state}}}) \tag{17}$$

where $W_{state} \in \mathbb{R}^{d \times \|V\|}$ with $V$ as the set of output vocabulary. As open-vocabulary DST models do not assume a known slot ontology, our models can generate the candidates from the dialogue history itself. To address OOV problem during inference, we incorporate a pointer network (Vinyals et al., 2015) into the Transformer decoder. We apply dot-product attention between the state decoder output and the stored memory of encoded dialogue history $Z$:

$$P_{\text{ptr}}^{\text{state}} = \text{softmax}(Z_{\text{ds} \times \text{fert}}^{T_{\text{state}}} Z^T) \tag{18}$$

The final probability of predicted state is defined as the weighted sum of the two probabilities:

$$P^{\text{state}} = p_{\text{gen}}^{\text{state}} \times P_{\text{vocab}}^{\text{state}} + (1 - p_{\text{gen}}^{\text{state}}) \times P_{\text{ptr}}^{\text{state}} \tag{19}$$

$$p_{\text{gen}}^{\text{state}} = \text{sigmoid}(W_{\text{gen}}V_{\text{gen}}) \tag{20}$$

$$V_{\text{gen}} = Z_{\text{ds} \times \text{fert}} \oplus Z_{\text{ds} \times \text{fert}}^{T_{\text{state}}} \oplus Z_{\text{exp}} \tag{21}$$

where $W_{\text{gen}} \in \mathbb{R}^{3d \times 1}$ and $Z_{\text{exp}}$ is the expanded vector of $Z$ to match dimensions of $Z_{\text{ds} \times \text{fert}}$. The final probability is used to train the state generation following the cross-entropy loss function:

$$\mathcal{L}_{\text{state}} = \sum_{d_g, s_h} \sum_{m=0}^{Y_f^{d_g, s_h}} -\log(P^{\text{state}}(y_m^{d_g, s_h})) \tag{22}$$

## 3.4 Optimization

We optimize all parameters by jointly training to minimize the weighted sum of the three losses:

$$\mathcal{L} = \mathcal{L}_{\text{state}} + \alpha \mathcal{L}_{\text{gate}} + \beta \mathcal{L}_{\text{fert}} \tag{23}$$

where $\alpha \geq 0$ and $\beta \geq 0$ are hyper-parameters.

## 4 Experiments

### 4.1 Dataset

MultiWOZ (Budzianowski et al., 2018) is one of the largest publicly available multi-domain task-oriented dialogue dataset with dialogue domains extended over 7 domains. In this paper, we use the new version of the MultiWOZ dataset published by Eric et al. (2019). The new version includes some correction on dialogue state annotation with more than 40% change across dialogue turns. On average, each dialogue has more than one domain. We pre-processed the dialogues by tokenizing, lower-casing, and delexicalizing all system responses following the pre-processing scripts from (Wu et al., 2019). We identify a total of 35 *(domain, slot)* pairs. Other details of data pre-processing procedures, corpus statistics, and list of *(domain, slot)* pairs are described in Appendix A.1.

### 4.2 Training Procedure

We use label smoothing (Szegedy et al., 2016) to train the prediction of dialogue state $Y$ but not for prediction of fertilities $Y_{\text{fert}}$ and gates $Y_{\text{gate}}$. During training, we adopt 100% teacher-forcing learning strategy by using the ground-truth of $X_{\text{ds}\times\text{fert}}$ as input to the state decoder. We also apply the same strategy to obtain delexicalized dialogue history i.e. dialogue history is delexicalized from the ground-truth belief state in previous dialogue turn rather than relying on the predicted belief state. During inference, we follow a similar strategy as (Lei et al., 2018) by generating dialogue state turn-by-turn and use the predicted belief state in turn $t-1$ to delexicalize dialogue history in turn $t$. During inference, $X_{\text{ds}\times\text{fert}}$ is also constructed by prediction $\hat{Y}_{\text{gate}}$ and $\hat{Y}_{\text{fert}}$. We adopt the Adam optimizer (Kingma & Ba, 2015) and the learning rate strategy similarly as (Vaswani et al., 2017). Best models are selected based on the best average joint accuracy of dialogue state prediction in the validation set. All parameters are randomly initialized with uniform distribution (Glorot & Bengio, 2010). We did not utilize any pretrained word- or character-based embedding weights. We tuned the hyper-parameters with grid-search over the validation set (Refer to Appendix A.2 for further details). We implemented our models using PyTorch (Paszke et al., 2017) and released the code on GitHub [1].

### 4.3 Baselines

The DST baselines can be divided into 2 groups: open-vocabulary approach and fixed-vocabulary approach as mentioned in Section 2. Fixed-vocabulary has the advantage of access to the known candidate set of each slot and has a high performance of prediction within this candidate set. However, during inference, the approach suffers from unseen slot values for slots with evolving candidates such as entity names and time- and location-related slots.

#### 4.3.1 Fixed-vocabulary

**GLAD** (Zhong et al., 2018). GLAD uses multiple self-attentive RNNs to learn a global tracker for shared parameters among slots and a local tracker for individual slot. The model utilizes previous system actions as input. The output is used to compute semantic similarity with ontology terms.

**GCE** (Nouri & Hosseini-Asl, 2018). GCE is a simplified and faster version of GLAD. The model removes slot-specific RNNs while maintaining competitive DST performance.

**MDBT** (Ramadan et al., 2018). MDBT model includes separate encoding modules for system utterances, user utterances, and *(slot, value)* pairs. Similar to GLAD, The model is trained based on the semantic similarity between utterances and ontology terms.

**FJST** and **HJST** (Eric et al., 2019). FJST refers to Flat Joint State Tracker, which consists of a dialog history encoder as a bidirectional LSTM network. The model also includes separate feedforward networks to encode hidden states of individual state slots. HJST follows a similar architecture but uses a hierarchical LSTM network (Serban et al., 2016) to encode the dialogue history.

**SUMBT** (Lee et al., 2019). SUMBT refers to Slot-independent Belief Tracker, consisting of a multi-head attention layer with query vector as a representation of a *(domain, slot)* pair and key and

---

[1]`https://github.com/henryhungle/NADST`

value vector as BERT-encoded dialogue history. The model follows a non-parametric approach as it is trained to minimize a score such as Euclidean distance between predicted and target slots. Our approach is different from SUMBT as we include attention among *(domain, slot)* pairs to explicitly learn dependencies among the pairs. Our models also generate slot values rather than relying on a fixed candidate set.

### 4.3.2 OPEN-VOCABULARY

**TSCP** (Lei et al., 2018). TSCP is an end-to-end dialogue model consisting of an RNN encoder and two RNN decoder with a pointer network. We choose this as a baseline because TSCP decodes dialogue state as a single sequence and hence, factor in potential dependencies among slots like our work. We adapt TSCP into multi-domain dialogues and report the performance of only the DST component rather than the end-to-end model. We also reported the performance of TSCP for two cases when the maximum length of dialogue state sequence $L$ in the state decoder is set to 8 or 20 tokens. Different from TSCP, our models dynamically learn the length of each state sequence as the sum of predicted fertilities and hence, do not rely on a fixed value of $L$.

**DST Reader** (Gao et al., 2019). DST Reader reformulates the DST task as a reading comprehension task. The prediction of each slot is a span over tokens within the dialogue history. The model follows an attention-based neural network architecture and combines a slot carryover prediction module and slot type prediction module.

**HyST** (Goel et al., 2019). HyST model combines both fixed-vocabulary and open-vocabulary approach by separately choosing which approach is more suitable for each slot. For the open-vocabulary approach, the slot candidates are formed as sets of all word n-grams in the dialogue history. The model makes use of encoder modules to encode user utterances and dialogue acts to represent the dialogue context.

**TRADE** (Wu et al., 2019). This is the current state-of-the-art model on the MultiWOZ2.0 and 2.1 datasets. TRADE is composed of a dialog history encoder, a slot gating module, and an RNN decoder with a pointer network for state generation. **SpanPtr** is a related baseline to TRADE as reported by Wu et al. (2019). The model makes use of a pointer network with index-based copying instead of a token-based copying mechanism.

### 4.4 RESULTS

We evaluate model performance by the joint goal accuracy as commonly used in DST (Henderson et al., 2014b). The metric compares the predicted dialogue states to the ground truth in each dialogue turn. A prediction is only correct if all the predicted values of all slots exactly match the corresponding ground truth labels. We ran our models for 5 times and reported the average results. For completion, we reported the results in both MultiWOZ 2.0 and 2.1.

As can be seen in Table 2, although our models are designed for non-autoregressive decoding, they can outperform state-of-the-art DST approaches that utilize autoregressive decoding such as (Wu et al., 2019). Our performance gain can be attributed to the model capability of learning cross-domain and cross-slot signals, directly optimizing towards the evaluation metric of joint goal accuracy rather than just the accuracy of individual slots. Following prior DST work, we reported the model performance on the *restaurant* domain in MultiWOZ 2.0 in Table 4. In this dialogue domain, our model surpasses other DST models in both Joint Accuracy and Slot Accuracy. Refer to Appendix A.3 for our model performance in other domains in both MultiWOZ2.0 and MultiWOZ2.1.

**Latency Analysis**. We reported the latency results in term of wall-clock time (in *ms*) per prediction state of our models and the two baselines TRADE (Wu et al., 2019) and TSCP (Lei et al., 2018) in Table 4. For TSCP, we reported the time cost only for the DST component instead of the end-to-end models. We conducted experiments with 2 cases of TSCP when the maximum output length of dialogue state sequence in the state decoder is set as $L = 8$ and $L = 20$. We varied our models for different values of $T = T_{\text{fert}} = T_{\text{state}} \in \{1, 2, 3\}$. All latency results are reported when running in a single identical GPU. As can be seen in Table 4, NADST obtains the best performance when $T = 3$. The model outperforms the baselines while taking much less time during inference. Our approach is similar to TSCP which also decodes a complete dialogue state sequence rather than individual slots to factor in dependencies among slot values. However, as TSCP models involve sequential

| Model | MultiWOZ2.1 | MultiWOZ2.0 |
|---|---|---|
| MDBT (Ramadan et al., 2018) [†] | - | 15.57% |
| SpanPtr (Vinyals et al., 2015) | - | 30.28% |
| GLAD (Zhong et al., 2018) [†] | - | 35.57% |
| GCE (Nouri & Hosseini-Asl, 2018) [†] | - | 36.27% |
| HJST (Eric et al., 2019) [*] | 35.55% | 38.40% |
| DST Reader (single) (Gao et al., 2019) [*] | 36.40% | 39.41% |
| DST Reader (ensemble) (Gao et al., 2019) | - | 42.12% |
| TSCP (Lei et al., 2018) | 37.12% | 39.24% |
| FJST (Eric et al., 2019) [*] | 38.00% | 40.20% |
| HyST (ensemble) (Goel et al., 2019) [*] | 38.10% | 44.24% |
| SUMBT (Lee et al., 2019) [†] | - | 46.65% |
| TRADE (Wu et al., 2019) [*] | 45.60% | 48.60% |
| **Ours** | **49.04%** | **50.52%** |

Table 2: DST Joint Accuracy metric on MultiWOZ 2.1 and 2.0. [†]: results reported on MultiWOZ2.0 leaderboard. [*]: results reported by Eric et al. (2019). Best results are highlighted in bold.

processing in both encoding and decoding, they require much higher latency. TRADE shortens the latency by separating the decoding process among *(domain, slot)* pairs. However, at the token level, TRADE models follow an auto-regressive process to decode individual slots and hence, result in higher average latency as compared to our approach. In NADST, the model latency is only affected by the number of attention layers in fertility decoder $T_{\text{fert}}$ and state decoder $T_{\text{state}}$. For approaches with sequential encoding and/or decoding such as TSCP and TRADE, the latency is affected by the length of source sequences (dialog history) and target sequence (dialog state). Refer to Appendix A.3 for visualization of model latency in terms of dialogue history length.

| Model | Joint Acc | Slot Acc |
|---|---|---|
| MDBT | 17.98% | 54.99% |
| SPanPtr | 49.12% | 87.89% |
| GLAD | 53.23% | 96.54% |
| GCE | 60.93% | 95.85% |
| TSCP | 62.01% | 97.32% |
| TRADE | 65.35% | 93.28% |
| **Ours** | **69.21%** | **98.84%** |

Table 3: DST joint accuracy and slot accuracy on MultiWOZ2.0 *restaurant* domain. Baseline results (except TSCP) were from Wu et al. (2019).

| Model | Joint Acc | Latency | Speed Up |
|---|---|---|---|
| TRADE | 45.60% | 362.15 | ×2.12 |
| TSCP (L=8) | 32.15% | 493.44 | ×1.56 |
| TSCP (L=20) | 37.12% | 767.57 | ×1.00 |
| **Ours (T=1)** | 42.98% | **15.18** | **×50.56** |
| **Ours (T=2)** | 45.78% | 21.67 | ×35.42 |
| **Ours (T=3)** | **49.04%** | 27.31 | ×28.11 |

Table 4: Latency analysis on MultiWOZ2.1. Latency is reported in terms of wall-clock time in *ms* per prediction state.

**Ablation Analysis**. We conduct an extensive ablation analysis with several variants of our models in Table 5. Besides the results of DST metrics, Joint Slot Accuracy and Slot Accuracy, we reported the performance of the fertility decoder in Joint Gate Accuracy and Joint Fertility Accuracy. These metrics are computed similarly as Joint Slot Accuracy in which the metrics are based on whether all predictions of gates or fertilities match the corresponding ground truth labels. We also reported the Oracle Joint Slot Accuracy and Slot Accuracy when the models are fed with ground truth $X_{\text{ds}\times\text{fert}}$ and $X_{\text{del}}$ labels instead of the model predictions. We noted that the model fails when positional encoding of $X_{\text{ds}\times\text{fert}}$ is removed before being passed to the state decoder. The performance drop can be explained because $PE$ is responsible for injecting sequential attributes to enable non-autoregressive decoding. Second, we also note a slight drop of performance when slot gating is removed as the models have to learn to predict a fertility of 1 for "none" and "dontcare" slots as well. Third, removing $X_{\text{del}}$ as an input reduces the model performance, mostly due to the sharp decrease in Joint Fertility Accuracy. Lastly, removing pointer generation and relying on only $P_{\text{vocab}}^{\text{state}}$ affects the model performance as the models are not able to infer slot values unseen during training, especially for slots such as *restaurant-name* and *train-arriveby*. We conduct other ablation experiments and report additional results in Appendix A.3.

| $X_{del}$ | Slot Gating | $PE(X_{ds \times fert})$ | Pointer Gen. | Joint Gate Acc | Joint Fert. Acc | Joint Slot Acc | Slot Acc | Oracle Joint Slot Acc | Oracle Slot Acc |
|---|---|---|---|---|---|---|---|---|---|
| ✓ | ✓ | ✓ | ✓ | **66.65%** | 63.18% | **49.04%** | 97.31% | **73.44%** | **99.01%** |
| ✓ | ✓ | | ✓ | 59.23% | 57.83% | 19.56% | 94.36% | 72.12% | 98.96% |
| ✓ | | ✓ | ✓ | N/A | **64.23%** | 48.74% | 96.62% | 73.01% | 98.97% |
| | ✓ | ✓ | ✓ | 48.23% | 45.35% | 39.45% | 95.92% | 66.27% | 98.63% |
| ✓ (no sys. act) | ✓ | ✓ | ✓ | 52.45% | 56.81% | 44.87% | 96.95% | 70.83% | 98.74% |
| ✓ | ✓ | ✓ | | 63.19% | 58.31% | 43.46% | 96.72% | 64.37% | 98.39% |
| | ✓ | ✓ | | 44.22% | 42.01% | 34.48% | 95.89% | 61.32% | 98.24% |
| | | ✓ | | N/A | 41.35% | 33.52% | 95.42% | 60.99% | 98.19% |

Table 5: Ablation analysis on MultiWOZ 2.1 on 4 components: partially delexicalized dialogue history $X_{del}$, slot gating, positional encoding $PE(X_{ds \times fert})$, and pointer network.

**Auto-regressive DST**. We conduct experiments that use an auto-regressive state decoder and keep other parts of the model the same. For the fertility decoder, we do not use Equation 14 and 16 as fertility becomes redundant in this case. We still use the output to predict slot gates. Similar to TRADE, we use the summation of embedding vectors of each domain and slot pair as input to the state decoder and generate slot value token by token. First, From Table 6, we note that the performance does not change significantly as compared to the non-autoregressive version. This reveals that our proposed NADST models can predict fertilities reasonably well and performance is comparable with the auto-regressive approach. Second, we observe that the auto-regressive models are less sensitive to the use of system action in dialogue history delexicalization. We expect this as predicting slot gates is easier than predicting fertilities. Finally, we note that our auto-regressive model variants still outperform the existing approaches. This could be due to the high-level dependencies among (domain, slot) pairs learned during the first part of the model to predict slot gates.

| MultiWOZ | Sys. Act | Joint Gate Acc | Joint Slot Acc | Slot Acc | Oracle Joint Slot Acc | Oracle Slot Acc |
|---|---|---|---|---|---|---|
| **2.1** | ✓ | 65.89% | 49.76% | 97.40% | 71.39% | 98.92% |
| **2.1** | | 62.04% | 46.57% | 97.23% | 66.72% | 98.65% |
| **2.0** | ✓ | 68.81% | 50.08% | 97.44% | 79.04% | 99.22% |
| **2.0** | | 65.27% | 50.46% | 97.43% | 76.21% | 99.08% |

Table 6: Performance of auto-regressive model variants on MultiWOZ2.0 and 2.1. Fertility prediction is removed as fertility becomes redudant in auto-regressive models.

**Visualization and Qualitative Evaluation**. In Figure 4, we include two examples of dialogue state prediction and the corresponding visualization of self-attention scores of $X_{ds \times fert}$ in state decoder. In each heatmap, the highlighted boxes express attention scores among non-symmetrical domain-slot pairs. In the first row, 5 attention heads capture the dependencies of two pairs *(train-leaveat, train-arriveby)* and *(train-departure, train-destination)*. The model prediction for these two slots matches the gold labels: *(train-leaveat, 09:50), (train-arriveby, 11:30)* and *(train-departure, cambridge), (train-destination, ely)* respectively. In the second row, besides slot-level dependency between domain-slot pairs *(taxi-departure, taxi-destination)*, token-level dependency is exhibited through the attention between *attraction-type* and *attraction-name*. By attending on token representations of *attraction-name* with corresponding output "christ college", the models can infer "attraction-type=college" correctly. In addition, our model also detects contextual dependency between *train-departure* and *attraction-name* to predict "train-departure=christ college." Refer to Appendix A.4 for the dialogue history with gold and prediction states of these two sample dialogues.

## 5 CONCLUSION

We proposed NADST, a novel Non-Autoregressive neural architecture for DST that allows the model to explicitly learn dependencies at both slot-level and token-level to improve the *joint* accuracy rather than just individual slot accuracy. Our approach also enables fast decoding of dialogue states by adopting a parallel decoding strategy in decoding components. Our extensive experiments on the well-known MultiWOZ corpus for large-scale multi-domain dialogue systems benchmark show that our NADST model achieved the state-of-the-art accuracy results for DST tasks, while enjoying a substantially low inference latency which is an order of magnitude lower than the prior work.

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

## A   APPENDIX

### A.1   DATASET PRE-PROCESSING

We follow similar data preprocessing procedures as Budzianowski et al. (2018) and Wu et al. (2019) on both MultiWOZ 2.0 and 2.1. The resulting corpus includes 8,438 multi-turn dialogues in training set with an average of 13.5 turns per dialogue. For the test and validation set, each includes 1,000 multi-turn dialogues with an average of 14.7 turns per dialogue. The average number of domains per dialogue is 1.8 for training, validation, and test sets. The MultiWOZ corpus includes much larger ontology than previous DST datasets such as WOZ (Wen et al., 2017) and DSTC2 (Henderson et al., 2014a). We identified a total of 35 *(domain, slot)* pairs across 7 domains. However, only 5 domains are included in the test data. Refer to Table 7 for the statistics of dialogues in these 5 domains.

| Domain | attraction | hotel | restaurant | taxi | train | All |
|---|---|---|---|---|---|---|
| **Slot** | area name type | area bookday bookpeople bookstay internet name parking pricerange stars type | area bookday bookpeople booktime food name pricerange | arriveby departure destination leaveat | arriveby bookpeople day departure destination leaveat | - |
| **train** | 3,381 | 3,103 | 2,717 | 3,813 | 1,654 | 8,438 |
| **val** | 416 | 484 | 401 | 438 | 207 | 1,000 |
| **test** | 394 | 494 | 395 | 437 | 195 | 1,000 |

Table 7: Summary of MultiWOZ dataset 2.1

### A.2   MODEL HYPER-PARAMETERS

We employed dropout (Srivastava et al., 2014) of 0.2 at all network layers except the linear layers of generation network components and pointer attention components. We used a batch size of 32, embedding dimension $d = 256$ in all experiments. We also fixed the number of attention heads to 16 in all attention layers. We shared the embedding weights to embed domain and slot tokens as input to fertility decoder and state decoder. We also shared the embedding weights between dialogue history encoder and state generator. We varied our models for different values of $T = T_{fert} = T_{state} \in \{1, 2, 3\}$. In all experiments, the warmup steps are fine-tuned from a range from 13K to 20K training steps.

### A.3   ADDITIONAL RESULTS

**Domain-specific Results**. We conduct experiments to evaluate our model performance in all 5 test domains in MultiWOZ2.0 and 2.1. From Table 8, our models perform better in *restaurant* and *attraction* domain in general. The performance in the *taxi* and *hotel* domain is significantly lower than other domains. This could be explained as the *hotel* domain has a complicated slot ontology with 10 different slots, larger than the other domains. For the *taxi* domain, we observed that dialogues with this domain are usually of multiple domains, including the *taxi* domain in combination with other domains. Hence, it is more challenging to track dialogue states in the *taxi* domain.

**Latency Results**. We visualized the model latency against the length of dialogue history in Figure 2 and 3. In Figure 2, we only plot with dialogue history length up to 80 tokens as TSCP models do not use the full dialogue history as input. In Figure 3, for a fair comparison between TRADE and NADST, we plot the latency of the original TRADE which decodes dialogue state slot by slot and a new version of TRADE* model which decodes individual slots following a parallel decoding mechanism. Since TRADE independently generates dialogue state slot by slot, we enable parallel generation simply by feeding all slots into models at once (without impacts on performance). However, at the token level, TRADE* still follows an autoregressive decoding framework. Compared to TRADE* and TSCP, our model latency is only dependent on the model complexity i.e. the number

|  | MultiWOZ2.1 | | MultiWOZ2.0 | |
|---|---|---|---|---|
| **Domain** | **Joint Acc** | **Slot** | **Joint Acc** | **Slot** |
| **Hotel** | 48.76% | 97.70% | 53.86% | 97.75% |
| **Train** | 62.36% | 98.36% | 58.58% | 98.08% |
| **Attraction** | 66.83% | 98.89% | 74.21% | 99.19% |
| **Restaurant** | 65.37% | 98.78% | 69.21% | 98.84% |
| **Taxi** | 33.80% | 96.69% | 34.94% | 96.76% |

Table 8: Additional domain-specific results of our model in MultiWOZ2.0 and MultiWOZ2.1. The model performs best with the *restaurant* domain and worst with the *taxi* domain.

of attention layers $T = T_{fert} = T_{state}$. For TRADE* and TSCP, the model latency increases as dialogue extends over time while NADST latency is almost constant. The non-constant latency is mostly due to overhead processing such as delexicalizing dialogue history. Our approach is, hence, suitable especially for dialogues in multiple domains as they usually extend over more number of turns (e.g. 13 to 14 turns per dialogue in average in MultiWOZ corpus) In Figure 3, we noted that the latency of the original TRADE is almost unchanged as the dialogue history extends. This is most likely due to the model having to decode all possible *(domain, slot)* pairs rather than just relevant pairs as in NADST and TSCP. The TRADE* shows a clearer increasing trend of latency because the parallel process is independent of the number of *(domain,slot)* pairs considered. TRADE* still requires more time to decode than NADST as we also parallelize decoding at the token level.

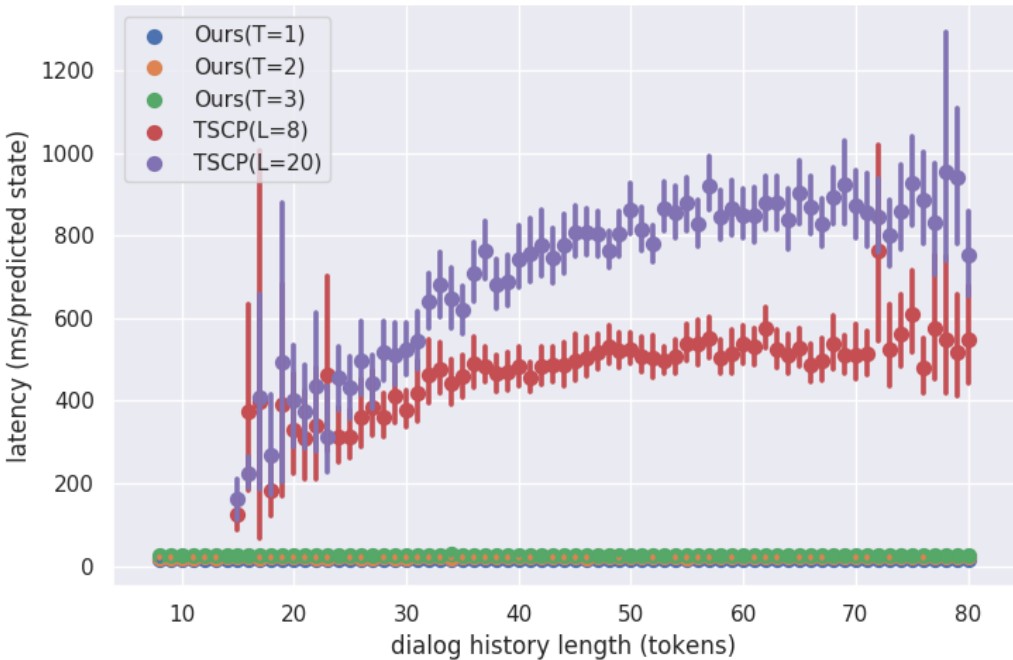

Figure 2: Comparison of model latency as wall-clock time (in *ms*) per prediction of complete dialogue state (not by individual slot). The latency is plotted against the length of the dialogue history. We compare our models with TSCP (Lei et al., 2018) with varied maximum output length of dialogue states $L = 8$ and $L = 20$. We vary our models with different values of number of attention layers $T = T_{fert} = T_{state} = 1, 2, 3$. Our models are more scalable as the latency does not change significantly when dialogue history extends over time.

**Ablation Results**. We conduct additional ablation experiments by varying the proportion of prediction values vs. ground-truth values for $X_{del}$ and $X_{ds \times fert}$ as input to the models. As can be seen in Table 9, the model performance increases gradually as the proportion of prediction input %pred reduces from 100% (true prediction) to 0% (oracle prediction). In particular, we observe more sig-

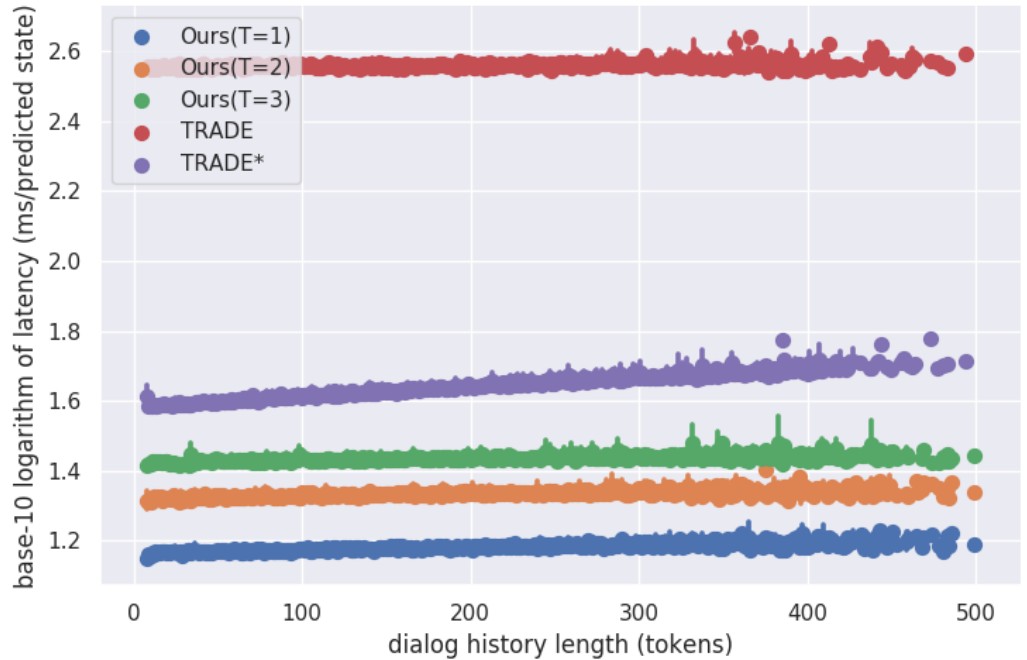

Figure 3: Comparison of model latency as wall-clock time (in *ms*) per prediction of complete dialogue state. The latency is plotted against the length of the dialogue history. For a fair comparison, we compare our models with TRADE (Wu et al., 2019) in 2 cases: original TRADE which decodes dialogue state slot by slot and TRADE* which decodes dialogue state in parallel at slot-level. Here we plot the base-10 logarithm of latency to show the difference between the 2 cases of TRADE. We vary our models with different values of number of attention layers $T = T_{fert} = T_{state} = 1, 2, 3$. Our models are more scalable as the latency does not change significantly when dialogue history extends over time.

nificant changes in performance against changes of %pred of $X_{ds \times fert}$. The model performance can increase up to more than 67% joint accuracy when we have an oracle input of $X_{ds \times fert}$. However, we consider improving model performance by $X_{del}$ more practically achievable. For example, we can make use of a more sophisticated mechanism to delexicalize dialog history rather than exact word matching as the current strategy. Another example is having better $X_{del}$ through a pretrained NLU model. In the ideal case with access to ground-truth labels of both $X_{del}$ and $X_{ds \times fert}$, the model can obtain a joint accuracy of 73%.

| %pred $X_{del}$ | %pred $X_{ds \times fert}$ | Joint Acc | Slot Acc | %pred $X_{del}$ | %pred $X_{ds \times fert}$ | Joint Acc | Slot Acc |
|---|---|---|---|---|---|---|---|
| 0% | 100% | 57.40% | 98.06% | 100% | 0% | 67.32% | 98.67% |
| 20% | 100% | 56.50% | 97.98% | 100% | 20% | 64.09% | 98.47% |
| 40% | 100% | 55.24% | 97.91% | 100% | 40% | 61.29% | 98.29% |
| 60% | 100% | 53.58% | 97.79% | 100% | 60% | 57.02% | 98.00% |
| 80% | 100% | 52.02% | 97.67% | 100% | 80% | 54.11% | 97.76% |
| 100% | 100% | 49.04% | 97.31% | 100% | 100% | 49.04% | 97.31% |
| 0% | 0% | 73.44% | 99.01% | 0% | 0% | 73.44% | 99.01% |

Table 9: Additional results of our model in MultiWOZ2.1 when we assume access to the ground-truth labels of $X_{del}$ and $X_{ds \times fert}$ (oracle prediction). We vary the the percentage of using the model prediction $\hat{X}_{del}$ and $\hat{X}_{ds \times fert}$ from 100% (true prediction) to 0% (oracle prediction).

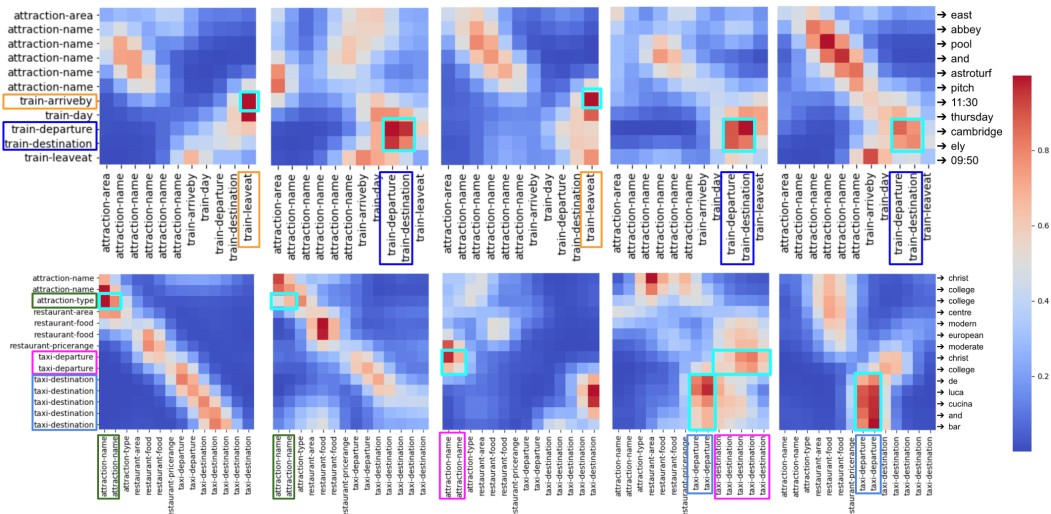

Figure 4: Heatmap visualization of self-attention scores of 5 heads between $Z_{ds \times fert}$ representations in the state decoder. The corresponding prediction output for each representation is presented on the right side. The examples are for the $6^{th}$ turn in dialogue ID MUL0536 (upper row) and PMUL3759 (lower row) in MultiWOZ2.1.

## A.4 SAMPLE PREDICTION OUTPUT

We extracted prediction output in all turns for 2 example dialogues: MUL0536 and PMUL3759.

| | | |
|---|---|---|
| **Turn 1** | **Human:** | i am looking for abbey pool and astroturf pitch can you help me ? |
| | **Gold Dialog State:** | (attraction-name, abbey pool and astroturf pitch) |
| | **Predicted Dialog State:** | (attraction-name, abbey pool and astroturf pitch) |
| **Turn 2** | **System:** | yes , abbey pool and astroturf pitch is a swimmingpool east of town . their number is 01223902088 , and address is pool way , whitehill road , off newmarket road . postcode cb58nt . |
| | **Human:** | thank you very much for the information . that is all i needed help with . have a nice day . |
| | **Gold Dialog State:** | (attraction-area, east), (attraction-name, abbey pool and astroturf pitch) , (attraction-type, swimming pool) |
| | **Predicted Dialog State:** | (attraction-area, east), (attraction-name, abbey pool and astroturf pitch) |
| **Turn 3** | **System:** | you are welcome . let me know if i can do anything else for you . |
| | **Human:** | i actually do need to find a train going to ely . |
| | **Gold Dialog State:** | (attraction-area, east), (attraction-name, abbey pool and astroturf pitch) , (attraction-type, swimming pool), (train-destination-ely) |
| | **Predicted Dialog State:** | (attraction-area, east), (attraction-name, abbey pool and astroturf pitch) , (train-destination-ely) |
| **Turn 4** | **System:** | is that leaving from cambridge ? and if so , what time would you like to arrive in ely ? |
| | **Human:** | yes , i will be leaving cambridge and going to ely , i would like it to arrive by 11:30 . |
| | **Gold Dialog State:** | (attraction-area, east), (attraction-name, abbey pool and astroturf pitch) , (attraction-type, swimming pool), (train-arriveby, 11:30), (train-departure, cambridge), (train-destination, ely) |
| | **Predicted Dialog State:** | (attraction-area, east), (attraction-name, abbey pool and astroturf pitch) , (train-arriveby, 11:30), (train-departure, cambridge), (train-destination, ely) |
| **Turn 5** | **System:** | what day would you be traveling ? there are 2,828 trains on that route . there is a train that departs friday at 9:50 and will arrive in ely at 10:07 . |
| | **Human:** | oops ! i guess forgot to mention it s thursday that i need to travel . |
| | **Gold Dialog State:** | (attraction-area, east), (attraction-name, abbey pool and astroturf pitch) , (attraction-type, swimming pool), (train-arriveby, 11:30), (train-day, thursday), (train-departure, cambridge), (train-destination, ely) |
| | **Predicted Dialog State:** | (attraction-area, east), (attraction-name, abbey pool and astroturf pitch) , (train-arriveby, 11:30), (train-day, thursday), (train-departure, cambridge), (train-destination, ely) |
| **Turn 6** | **System:** | there are 3 trains that would fit , leaving at 05:50 , 07:50 , or 09:50 . |
| | **Human:** | can i get info for the 09:50 the price and the trains id please ? |
| | **Gold Dialog State:** | (attraction-area, east), (attraction-name, abbey pool and astroturf pitch) , (attraction-type, swimming pool), (train-arriveby, 11:30), (train-day, thursday), (train-departure, cambridge), (train-destination, ely), (train-leaveat, 09:50) |
| | **Predicted Dialog State:** | (attraction-area, east), (attraction-name, abbey pool and astroturf pitch) , (train-arriveby, 11:30), (train-day, thursday), (train-departure, cambridge), (train-destination, ely), (train-leaveat, 09:50) |
| **Turn 7** | **System:** | certainly . the train s id is tr1923 , and the price for a ticket is 4.40 pounds . |
| | **Human:** | great , thank you ! that will be all i need for now . |
| | **Gold Dialog State:** | (attraction-area, east), (attraction-name, abbey pool and astroturf pitch) , (attraction-type, swimming pool), (train-arriveby, 11:30), (train-day, thursday), (train-departure, cambridge), (train-destination, ely), (train-leaveat, 09:50) |
| | **Predicted Dialog State:** | (attraction-area, east), (attraction-name, abbey pool and astroturf pitch) , (train-arriveby, 11:30), (train-day, thursday), (train-departure, cambridge), (train-destination, ely), (train-leaveat, 09:50) |
| **Turn 8** | **System:** | are you certain you do not need further assistance ? |
| | **Human:** | 9:50 departure , 4.40 pounds , tr1923 . i got it , thank you ! |
| | **Gold Dialog State:** | (attraction-area, east), (attraction-name, abbey pool and astroturf pitch) , (attraction-type, swimming pool), (train-arriveby, 11:30), (train-day, thursday), (train-departure, cambridge), (train-destination, ely), (train-leaveat, 09:50) |
| | **Predicted Dialog State:** | (attraction-area, east), (attraction-name, abbey pool and astroturf pitch) , (train-arriveby, 11:30), (train-day, thursday), (train-departure, cambridge), (train-destination, ely), (train-leaveat, 09:50) |

Table 10: Full set of predicted dialogue states for dialogue ID MUL0536 in MultiWOZ2.1.

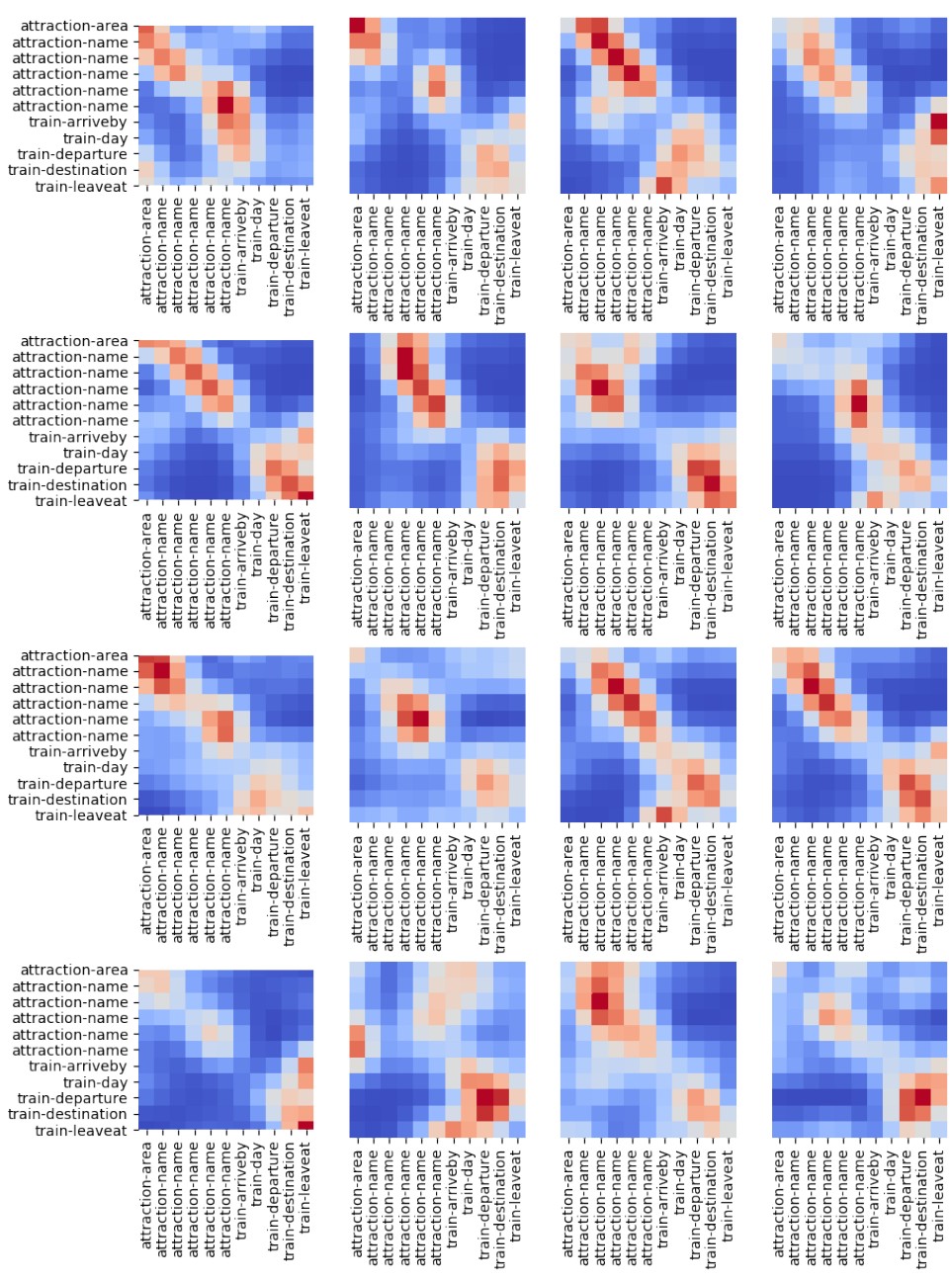

Figure 5: Visualization of all attention heads in the last attention step $T_{state}$ in the state decoder. The DST prediction is done for the $6^{th}$ turn in dialogue ID MUL0536 in MultiWOZ2.1.

| | | |
|---|---|---|
| **Turn 1** | **Human:** | what s your favorite college to visit in cambridge ? |
| | **Gold Dialog State:** | (attraction-type, college) |
| | **Predicted Dialog State:** | (attraction-type, college) |
| **Turn 2** | **System:** | i like christ s college in the center of town , but there are many others throughout the city . what part of the city did you want to visit ? |
| | **Human:** | i think i would like to visit that location , it seems nice . could i get the phone number and the area ? |
| | **Gold Dialog State:** | (attraction-name, christ college), (attraction-type, college) |
| | **Predicted Dialog State:** | (attraction-area, centre), (attraction-name, christ college), (attraction-type, college) |
| **Turn 3** | **System:** | the area is centre and the phone number is 01223334900 . is there anything else i can help you with ? |
| | **Human:** | actually , i am hungry . can you find me a restaurant that serves north american food ? something not super expensive , maybe in a moderate price range ? |
| | **Gold Dialog State:** | (attraction-name, christ college), (attraction-type, college), (restaurant-food, modern american), (restaurant-pricerange, moderate) |
| | **Predicted Dialog State:** | (attraction-name, christ college), (attraction-type, college), (restaurant-food, north american), (restaurant-pricerange, moderate) |
| **Turn 4** | **System:** | i am sorry , there s no restaurant serving specifically north american or american food in my database , is there another type of food you would consider ? |
| | **Human:** | how about modern european food ? |
| | **Gold Dialog State:** | (attraction-name, christ college), (attraction-type, college), (restaurant-food, modern european), (restaurant-pricerange, moderate) |
| | **Predicted Dialog State:** | (attraction-name, christ college), (attraction-type, college), (restaurant-food, modern european), (restaurant-pricerange, moderate) |
| **Turn 5** | **System:** | there are 3 modern european restaurant -s 2 in the center and 1 in the south . do you have a preference ? |
| | **Human:** | i would prefer the 1 on the centre , could i have the phone number and postcode please ? |
| | **Gold Dialog State:** | (attraction-name, christ college), (attraction-type, college), (restaurant-area, centre), (restaurant-food, modern european), (restaurant-pricerange, moderate) |
| | **Predicted Dialog State:** | (attraction-name, christ college), (attraction-type, college), (restaurant-area, centre), (restaurant-food, modern european), (restaurant-pricerange, moderate) |
| **Turn 6** | **System:** | de luca cucina and bar s phone number is 01223356666 . postcode is cb21aw . |
| | **Human:** | could you help me get a taxi to get from the college to the restaurant ? |
| | **Gold Dialog State:** | (attraction-name, christ college), (attraction-type, college), (restaurant-area, centre), (restaurant-food, modern european), (restaurant-pricerange, moderate), (taxi-departure, christ college), (taxi-destination, de luca cucina and bar) |
| | **Predicted Dialog State:** | (attraction-name, christ college), (attraction-type, college), (restaurant-area, centre), (restaurant-food, modern european), (restaurant-pricerange, moderate), (taxi-departure, christ college), (taxi-destination, de luca cucina and bar) |
| **Turn 7** | **System:** | what time would you like to leave the college ? i can book you a taxi to take you to the restaurant if you would like . |
| | **Human:** | i would like to leave by 13:00 . |
| | **Gold Dialog State:** | (attraction-name, christ college), (attraction-type, college), (restaurant-area, centre), (restaurant-food, modern european), (restaurant-pricerange, moderate), (taxi-departure, christ college), (taxi-destination, de luca cucina and bar), (taxi-leaveat, 12:45) |
| | **Predicted Dialog State:** | (attraction-name, christ college), (attraction-type, college), (restaurant-area, centre), (restaurant-food, modern european), (restaurant-pricerange, moderate), (taxi-departure, christ college), (taxi-destination, de luca cucina and bar), (taxi-leaveat, 13:00) |
| **Turn 8** | **System:** | i have booked you a taxi leaving at 12:45 . the car will be a red toyota and contact number is 07350032543 . anything else today ? |
| | **Human:** | that s it . thank you very much . |
| | **Gold Dialog State:** | (attraction-name, christ college), (attraction-type, college), (restaurant-area, centre), (restaurant-food, modern european), (restaurant-pricerange, moderate), (taxi-departure, christ college), (taxi-destination, de luca cucina and bar), (taxi-leaveat, 12:45) |
| | **Predicted Dialog State:** | (attraction-name, christ college), (attraction-type, college), (restaurant-area, centre), (restaurant-food, modern european), (restaurant-pricerange, moderate), (taxi-departure, christ college), (taxi-destination, de luca cucina and bar), (taxi-leaveat, 12:45) |
| **Turn 9** | **System:** | will you need anymore information concerning your stay ? |
| | **Human:** | that is all , thanks for the help . |
| | **Gold Dialog State:** | (attraction-name, christ college), (attraction-type, college), (restaurant-area, centre), (restaurant-food, modern european), (restaurant-pricerange, moderate), (taxi-departure, christ college), (taxi-destination, de luca cucina and bar), (taxi-leaveat, 12:45) |
| | **Predicted Dialog State:** | (attraction-name, christ college), (attraction-type, college), (restaurant-area, centre), (restaurant-food, modern european), (restaurant-pricerange, moderate), (taxi-departure, christ college), (taxi-destination, de luca cucina and bar), (taxi-leaveat, 12:45) |

Table 11: Full set of predicted dialogue states for dialogue ID PMUL3759 in MultiWOZ2.1.

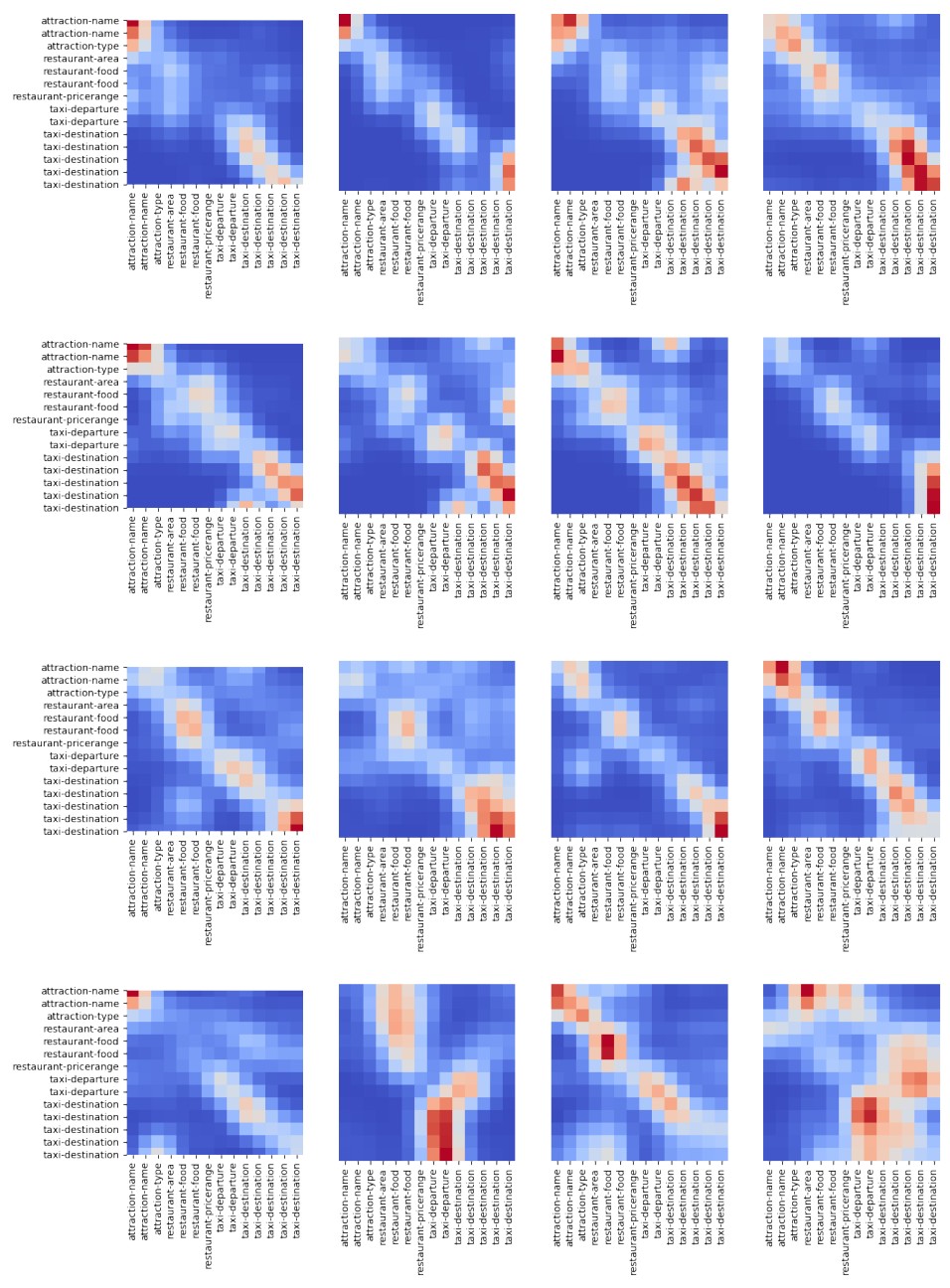

Figure 6: Visualization of all attention heads in the last attention step $T_state$ in the state decoder. The DST prediction is done for the $6^{th}$ turn in dialogue ID PMUL3759 in MultiWOZ2.1.

