# OpenReview forum: "Non-Autoregressive Dialog State Tracking"
_ICLR.cc/2020/Conference — Accept (Poster)_

### Official Review · AnonReviewer2 · 2019-10-23
**Official Blind Review #2**

**Rating:** 6

**Review:**

[Contribution summary]
Authors propose a new model for the DST task that (1) reduces the inference time complexity with an non-autoregressive decoder, and (2) obtains the competitive DST accuracy (49.04% joint accuracy on MultiWoZ 2.1).

[Comments]
- The proposed model is well motivated and well structured. Empirical results show improvement over other baselines, with the main gain coming from delexicalization, slot gating, fertility output, etc.

- Some of the details are not entirely provided - e.g. please provide the loss hyper-parameter values (e.g. Eq.23) and optimizer parameters for the training.

- Overall presentation, notations, figures, etc. could improve.

- There have been recent work on DST with new SOTA results (e.g. “Towards Scalable Multi-domain Conversational Agents: The Schema-Guided Dialogue Dataset” by Rastogi et al.) -- please consider comparing the approaches.


**Experience Assessment:**

I have read many papers in this area.

**Review Assessment: Checking Correctness Of Derivations And Theory:**

I carefully checked the derivations and theory.

**Review Assessment: Checking Correctness Of Experiments:**

I carefully checked the experiments.

**Review Assessment: Thoroughness In Paper Reading:**

I read the paper thoroughly.

---

> ### Author Response · Authors · 2019-11-15
> **Thanks for the positive comments and the concerns!**
>
> Thank you very much for your review. Below are our responses:
>
> 1. About the details of parameter settings, we simply set the loss hyper-parameters $\alpha$ and $\beta$ to 1.  We set the optimizer parameters for training to $\beta_1=0.9$, $\beta_2=0.98$, and $\epsilon=10^{-9}$.
>
> 2. Thanks for pointing out the presentation issue. We made some improvement including standardizing the notations. We will further improve the paper presentation in the final version.
>
> 3. Thanks for pointing out the new paper. We note that the paper addresses a different setting and the authors employed a different evaluation metric instead of traditional joint goal accuracy. This metric is based on a Fuzzy Matching score on non-categorical slots but traditional metric uses exact word matching instead.  We are not sure if this metric is the one reported in the paper. We also do not know how the authors decide which slot is categorical or non-categorical in the MultiWOZ benchmark. Therefore, it is difficult for us to make a direct comparison of our model performance to this work.

---

### Official Review · AnonReviewer3 · 2019-10-23
**Official Blind Review #3**

**Rating:** 1

**Review:**

This paper proposed a model that is capable of tracking dialogue states in a non-recursive fashion. The main techniques behind the non-recursive model is similar to that of the ICLR 2018 paper "NON-AUTOREGRESSIVE NEURAL MACHINE TRANSLATION". Unfortunately, as state tacking can be formulated as one special case of sequence decoding, there is not much of innovation that can be claimed in this paper considering the "fertility" idea was already been proposed. The paper did illustrate a strong experimental results on a recent dataset comparing with many state-of-the-art models. However, it is not clear how much innovation this work generates and how the ICLR community would benefit from the problem that the paper is addressing.



**Experience Assessment:**

I have read many papers in this area.

**Review Assessment: Checking Correctness Of Derivations And Theory:**

I assessed the sensibility of the derivations and theory.

**Review Assessment: Checking Correctness Of Experiments:**

I carefully checked the experiments.

**Review Assessment: Thoroughness In Paper Reading:**

I read the paper at least twice and used my best judgement in assessing the paper.

---

> ### Author Response · Authors · 2019-11-15
> **Thanks for your review! We clarify and highlight our contributions.**
>
> Thanks for reviewing our work. Below are our responses to clarify a few misunderstanding points by the reviewer.
>
> 1. We note that NMT and DST are very different research tasks, and DST has different settings, unique characteristics and challenges. NMT aims to generate an output sequence that has the same meaning as input sequence and the two sequences are of different languages/domains. By contract, DST aims to make accurate dialogue state prediction, instead of matching the semantic meanings between input and output. Our DST approach treats input as a prior for each (domain, slot) pair, and the prior is used as a channel vector to obtain high-level and low-level (token-level) dependencies to achieve better dialogue state prediction. Hence, the proposed technical approach and contributions in our work are actually quite different from the previous NMT work, although the idea of non-autoregressive was partly inspired by ICLR’18 NMT paper.
>
> 2. About the novelty and innovation, we highlight that one of our key contributions is to overcome a critical limitation of many existing DST methods. These methods do not detect dependencies among (domain, slot) pairs because they do not allow the model to explicitly learn signals across domains and slots. This is a very important contribution that is unique to DST tasks, and is very different from the previous non-autoregressive NMT work. We also noted that technically it is non-trivial and not straightforward at all to apply the non-autoregressive idea to develop a new state-of-the-art DST technique, which involves several other techniques (such as delexicalization, designs of slot gating, fertility, etc as presented in our paper).
>
> 3. About the significance of this work, our new approach not only achieves the new state-of-the-art results for DST tasks on the MultiWOZ dataset, but the decoding speed is also an order of magnitude faster than the existing DST methods, making it a practically very useful technique to many real-world dialog systems applications. From a scientific aspect, our technique is also not restricted to DST tasks. It potentially could be applied to tackle many other similar NLP or machine learning problems that involve structured or compositional prediction such as outline-based or template-based generation.

---

### Official Review · AnonReviewer1 · 2019-10-30
**Official Blind Review #1**

**Rating:** 6

**Review:**

The authors build on recent work for non-autoregressive encoder-decoder models in the context of machine translation (most significantly [Gu, et al., ICLR18]) and adapt this to dialogue state tracking. Specifically, as in [Gu, et al, ICLR18], they use a fertility decoder modified for DST to be on a per-slot basis which is input to a second decoder to generate the (open-vocabulary) tokens representing dialogue state. An interesting aspect of the resulting model as formulated is that the latent space also takes into account interdependencies between generated slot values, which leads to a direct structured prediction-like loss of joint accuracy. Additionally, as slot values have a smaller (and likely peakier) combinatorial space, NAT models actually are more applicable to DST than MT. The resulting model achieves state-of-the-art empirical results on the MultiWOZ dataset while incurring decoding times that are an order of magnitude faster.

Once one understands this conceptual modification of modifying the NAT string encoder-decoder to a more structured NAT encoder-decoder (which in DST is more of a normalized string), they apply all of the state-of-the-art techniques to build a DST system: gating of all potential (domain, slot) pairs as an auxiliary prediction (e.g., [Wu et al., ACL19]), de-lexicalizing defined value types [Lei, et al., ACL18], content encoder, and domain-slot encoder (with pretty standard hyper-parameters, etc.). The significant addition is the fertility decoder and the associated NAT state decoder. Thus, from a conceptual level, this isn’t a huge leap and something many researchers *could* have done (i.e., I think many people, including myself, to have expected this paper to come out) — thus, it is more of a ‘done first’ paper than a ‘done unexpectedly’ paper. However, it is done well and the results are convincing and interesting. Given the impressive performance, I expect others to continue building on this work and potentially even influencing people to combine encoder-decoder models with a more structured prediction approach to DST.  Thus, I would prefer to see it accepted if possible.

That being said, I do have a few questions regarding this work — but these are more questions that might be able to be addressed than actual criticisms per se. First, in Table 5, why without the delexicalized dialogue history does the performance drop from 49.04% to 39.45%? This does not make sense to me as the model is much more complex than TRADE; however, TRADE does not do delexicalization yet achieves 45.6% joint accuracy. Meanwhile, with such complex model, I would expect the model can learn from raw data without delexicalization. Moreover, the proposed method use both previous predictions and *previous system actions* to do delexicalization. Also, the NATMT models don’t do delexicalization (although they have significantly more data). I think the authors should do an ablation study that do not use previous system actions, because this is extra information compared with TRADE — even if delexicalizing. Secondly, another worthy baseline would be an autoregressive decoder (with other blocks of the model remain the same). I’d assume that the performance is slight higher. It is interesting to see the gap, because this gap is the cost to speed up decoder using fertility — even if it is a bit counter-intuitive. If there is no improvement in this setting, then structured prediction in general may make more sense. Honestly, I think more would be interested in the second point than the first.

In any case, nice paper — well-written, well-motivated, interesting empirical results. The only reason I am recommending ‘weak accept’ is that the novelty is fairly straightforward and the strength is in the execution.

**Experience Assessment:**

I have published one or two papers in this area.

**Review Assessment: Checking Correctness Of Derivations And Theory:**

N/A

**Review Assessment: Checking Correctness Of Experiments:**

I carefully checked the experiments.

**Review Assessment: Thoroughness In Paper Reading:**

I read the paper thoroughly.

---

> ### Author Response · Authors · 2019-11-15
> **We conducted additional experiments to clarify our model performance.**
>
> Thank you very much for your informative and constructive review. Please find our responses to your questions below.
>
> 1. About the performance drop without the delexicalized dialogue history in Table 5, it is caused mostly due to the performance decrease of the fertility decoder.  As shown in Table 5, we can see the joint gate accuracy and joint fertility accuracy reduce significantly from 66.7% to 48.2% and 63.2% to 45.4% respectively. This leads to error cascading to our state decoder because many of $X_\mathrm{ds \times fert}$ input is not correctly defined.  We consider using delexicalization because the fertility prediction is a challenging task and delexicalized dialogue history can improve this part of the model and hence, maintain competitive performance with current auto-regressive approaches.
>
> 2. Per your suggestion, we conducted an additional experiment on MultiWOZ 2.1 without using previous system action to delexicalize dialogue history. From the updated results in Table 5, we noted that the performance drops to 44.9%.  As mentioned in point [1], the performance drop is caused by the impact on fertility prediction which has the accuracy drop to 56.8% in this case.  We consider the use of system action and delexicalized dialogue history to improve the performance of fertility prediction and consequentially the state prediction. For fair comparison with TRADE which is auto-regressive model, we conduct experiments with auto-regressive version of our models (see point [3] below) with or without using system actions in Table 6. With this version, our model still achieves the SOTA (50.1% in MultiWOZ2.0 and 46.6% in MultiWOZ2.1) which could be due to the high-level dependencies learned among (domain, slot) pairs in fertility decoder.
>
> 3. Per your suggestion, we implemented a new baseline by replacing the current state decoder with an autoregressive decoder in the proposed NADST. As fertilities are redundant in an autoregressive model, we keep the same architecture but only use the output to predict slot gates.  In the state decoder, value of each (domain, slot) pair is decoded by sequentially passing previously generated tokens as input to the decoder.  Similar to TRADE, the first input token to the decoder is the summed of the corresponding domain and slot representations and the output sequence is decoded greedily. From the updated results in Table 6, we can observe the comparable performance as compared with our non-autoregressive version (50.5% vs. 50.1% in MultiWOZ2.0 and 49.0% vs. 49.8% in MultiWOZ2.1).  This shows that our NADST models can predict fertilities and slot gates reasonably well and therefore decode all slot values altogether in a non-autoregressive manner without degrading the performance.  Other observations are also stated in detail in our paper.

---

### Public Comment · ~Shuai_Lin1 · 2019-10-09
**Some questions about the notations in this paper**

Thanks for this nice work! Now I have some questions about this paper:

1.Is the equation (9) correct? I think the 'V' should be concluded by the parentheses.
2.Is the notation of 'Z^(ds)'  in equation (10~12)  correct? The meaning of this should be the set of the context vectors. What's  the difference between it with the 'Z^(ds) in equation (6)?
3.How to obtain the ground-truth of the fertilities when computing the cross-entropy loss in equation (16)?
4.Why is the reported result of SUMBT in Table 2 different from the one in the original paper ?
5.Do you plan to release the code? Otherwise, it may seem to be hard to reproduce .

---

> ### Author Response · Authors · 2019-10-15
> **Clarification**
>
> Thank you very much for your interest in our work. Please find below our response:
> 1. Yes this is a writing mistake. The $V$ should be concluded by the parentheses. We will modify this in the revision.
> 2. We agree that the notation could be improved in these equations. The output $Z_{ds}$ in equation (10-12) is the attended vector output from the domain-slot self-attention, delexicalized context attention, and context attention respectively.  To differentiate it from $Z_{ds}$, we can denote $Z^{out}_{ds}$ as output in equation (10-12):
> $$
> Z^{out}_{ds} = Attention(Z_{ds}, Z_{ds}, Z_{ds})
> $$
> $$
> Z^{out}_{ds} = Attention(Z^{out}_{ds}, Z_{del}, Z_{del})
> $$
> $$
> Z^{out}_{ds} = Attention(Z^{out}_{ds}, Z, Z)
> $$
> 3. The ground-truth fertilities are computed as token-level length of slot values in a dialogue state. For example, a dialogue state "(attraction-area, centre), attraction-name, the cambridge corn exchange)" has the ground-truth fertility for attration-area slot equal to 1 and fertility for attraction-name slot equal to 4. All other slots will have the default fertility of 0.
> 4. We reported the SUMBT result as shown in the MultiWoZ leaderboard (http://dialogue.mi.eng.cam.ac.uk/index.php/corpus/). We noted this result is higher than the one reported in the original paper.
> 5. Yes we plan to release the code.

---

### Decision · Program_Chairs · 2019-12-19

**Decision:**

Accept (Poster)

**Comment:**

(Please note that I am basing the meta-review on two reviews plus my own thorough read of the paper)
This paper proposes an interesting adaptation of the non-autoregressive neural encoder-decoder models previously proposed for machine translation to dialog state tracking. Experimental results demonstrate state-of-the-art for the MultiWOZ, multi-domain dialog corpus. The reviewers suggest that while the NA approach is not novel, author's adaptation of the approach to dialog state tracking and detailed experimental analysis are interesting and convincing. Hence I suggest accepting the paper as a poster presentation.